# Highly Sensitive Capacitive MEMS for Photoacoustic Gas Trace Detection

**DOI:** 10.3390/s23063280

**Published:** 2023-03-20

**Authors:** Tarek Seoudi, Julien Charensol, Wioletta Trzpil, Fanny Pages, Diba Ayache, Roman Rousseau, Aurore Vicet, Michael Bahriz

**Affiliations:** IES, CNRS, University of Montpellier, 34095 Montpellier, France; tarek.seoudi@umontpellier.fr (T.S.); julien.charensol@umontpellier.fr (J.C.); wioletta.trzpil@cea.fr (W.T.); fanny.pages@etu.umontpellier.fr (F.P.); diba.ayache@ies.univ-montp2.fr (D.A.); roman.rousseau@ec-lyon.fr (R.R.); aurore.vicet@umontpellier.fr (A.V.)

**Keywords:** photoacoustic spectroscopy, gas sensor, micro-mechanical resonator, MEMS, capacitive transduction

## Abstract

An enhanced MEMS capacitive sensor is developed for photoacoustic gas detection. This work attempts to address the lack of the literature regarding integrated and compact silicon-based photoacoustic gas sensors. The proposed mechanical resonator combines the advantages of silicon technology used in MEMS microphones and the high-quality factor, characteristic of quartz tuning fork (QTF). The suggested design focuses on a functional partitioning of the structure to simultaneously enhance the collection of the photoacoustic energy, overcome viscous damping, and provide high nominal capacitance. The sensor is modeled and fabricated using silicon-on-insulator (SOI) wafers. First, an electrical characterization is performed to evaluate the resonator frequency response and nominal capacitance. Then, under photoacoustic excitation and without using an acoustic cavity, the viability and the linearity of the sensor are demonstrated by performing measurements on calibrated concentrations of methane in dry nitrogen. In the first harmonic detection, the limit of detection (LOD) is 104 ppmv (for 1 s integration time), leading to a normalized noise equivalent absorption coefficient (NNEA) of 8.6 ⋅ 10^−8^ Wcm^−1^ Hz^−1/2^, which is better than that of bare Quartz-Enhanced Photoacoustic Spectroscopy (QEPAS), a state-of-the-art reference to compact and selective gas sensors.

## 1. Introduction

Gas sensors play a crucial role to control and monitor industrial and urban gas emissions. For applications related to human health like breath analysis or the detection of toxic/explosive gas such as methane, there is a need for a highly sensitive, selective, and compact gas sensor. The miniaturization of gas sensor is not only cost-effective, but also allows portability, which makes it suitable for most applications. Over the last decade, several gas sensors have been developed to detect and analyze a wide range of gases. The most commonly used gas sensors are electrochemical, semiconductor, and infrared (IR) sensors [1]. Electrochemical and semiconductor sensors are characterized by their compactness and relatively low cost. Electrochemical sensors detect electron transfer events during electrochemical reactions, while semiconductor sensors operate by measuring changes in electrical resistance caused by the adsorption of target gas molecules. Both types of sensors are sensitive to the detection of gases such as methane, where the limit of detection can reach less than 9 ppmv and 5 ppmv for electrochemical and semiconductors sensors, respectively [2,3]. However, these sensors have decreased reliability and limited lifetime due to the impact of humidity and temperature, as well as the gradual consumption associated with chemical reactions. In addition, cross-sensitivity can occur for some gases, making these gas sensors less selective.

Among the various detection techniques, tunable laser diode spectrometry (TDLS) offers excellent properties in terms of selectivity, reliability, and sensitivity. It is also capable of reaching ppb levels, as illustrated in [4], for methane detection. This technique includes a semiconductor modulated infrared laser whose wavelength corresponds to the absorption line of the target gas. This technique ensures a high selectivity due to the fact that the laser line is much thinner than the gas absorption line. The detection consists in measuring the intensity of the laser light passing through the gas cell using a photodiode. According to the Beer–Lambert law, the sensitivity of the IR absorption technique is proportional to the length of the absorption path, meaning that the performance relies on the size of the sensor. Therefore, the lack of compactness represents the main drawback of such a technique. This issue can be overcome by using photoacoustic spectroscopy which, compared to TDLS, differs in the way the light absorption is measured. After the absorption of the laser light by the targeted gas, the excited gas molecules will return into their fundamental level of energy after a non-radiative transition. This will increase the kinetic energy of the molecules and generate a periodic heating of the medium at the same frequency of the laser modulation, giving rise to acoustic waves having the same period. Hence, rather than using photodetectors, the measurement is performed using an acoustic sensor. In this case, the sensitivity is not proportional to the optical path, but to the power of the laser, which allows the development of compact gas sensors working with a small gas volume.

Each photoacoustic technique is defined by its acoustic sensor. The most used are

MPAS (Microphone based Photoacoustic Spectroscopy), which uses microphones with capacitive transduction;CEPAS (Cantilever-Enhanced Photoacoustic Spectroscopy), which uses cantilevers with optical transduction;QEPAS (Quartz-Enhanced Photoacoustic Spectroscopy), which uses quartz tuning forks (QTF) with piezoelectric transduction.

In MPAS, the main drawback is the interference of the ambient background with measurements [5]. As for CEPAS, although it is efficient in terms of sensitivity, the issue lies in its sizable optical readout mechanism and in the need for an acoustic cavity [6]. Regarding the bare QEPAS, it suffers from poor spatial overlapping between the acoustic wave and the prongs of the tuning fork, which induces a reduction in the signal-to-noise ratio (SNR) [7].

These different drawbacks can be explained by the fact that none of these acoustic sensors are fully designed for photoacoustic spectroscopy. In this context, an innovative approach of developing silicon-based sensor, compact and integrated like a microphone and clearly resonant like a QTF, is adopted. We initially proposed this technique, called MEMPAS (MEMS-Enhanced Photoacoustic Spectroscopy) in 2019 [8]. It consists of using a capacitive silicon micro-mechanical resonator, fabricated on a silicon-on-insulator (SOI) wafer. The use of silicon has the advantages of mature CMOS technology, notably precise dimension control, integration, and cost reduction for batch production. In addition, capacitive sensing is an attractive option which offers construction simplicity; highly doped silicon can be used as a capacitive electrode instead of having to perform metal deposition. An ideal capacitor has no intrinsic noise and is thus suitable for low noise sensing [9]. These features allow for a sensor structure specifically designed for photoacoustic gas sensing. Evidence shows that it offers a high quality factor compared to microphones [8].

However, the performance achieved by the first prototype we developed was far from the state of the art [8]. This can be explained by the capacitive transduction mechanism, based on gap reduction, which generates an important viscous squeeze film effect due to the proximity between the two electrodes as described in [10]. This effect reduces the mechanical resonator displacement; thus, the capacitance variations is limited and, therefore, its sensitivity. To minimize this effect, the resonator surface should be decreased. However, this reduces the collection of photoacoustic energy, which means that capacitive transduction exhibits opposite physical trends. This occurs because the same part fulfills two functions: photoacoustic energy collection and capacitive transduction. To overcome this problem, we have proposed an innovative design, named H-resonator [11], which allocates these two functions to two distinct parts. Although the H-resonator exhibits performances close to the bare QEPAS technique, achieving superior sensitivity appears feasible by improving the photoacoustic energy collection, the displacement of the movable electrodes, and the nominal capacitance. In this regard, this study focuses on improving the H-resonator. The goal is to develop highly sensitive capacitive MEMS for photoacoustic gas trace detection. In this work, an electrical characterization is performed to determine the resonator frequency response and nominal capacitance. Then, under photoacoustic excitation, without using an acoustic cavity, the linearity of the sensor is examined under calibrated concentrations of the target gas. Finally, the performances of the developed sensor are evaluated in terms of its limit of detection (LOD) and its Normalized Noise Equivalent Absorption (NNEA).

## 2. Silicon Micro-Mechanical Resonator Design

The mechanical resonator is fabricated on a double-side polish (100) oriented SOI wafer. This latter is composed of 3 main layers: a 75-µm thick silicon device layer which corresponds to the resonator thickness, a 3 µm-thick box layer composed of silicon oxide, and a 400 µm-thick silicon substrate layer. The device and substrate layers are made of highly boron doped silicon with a resistivity of 0.01~0.02 Ω·cm to avoid any metal deposition and ensure capacitive transduction.

Figure 1 illustrates the proposed mechanical resonator, which will be referred to as the H-square resonator. It is an evolution of the previous design, the H-resonator [11]. It is based on the same working principle that is presented in Figure 1. The acoustic wave is created along the beam due to the absorption of modulated light. This absorption will occur if the laser wavelength corresponds to the absorption line of a target gas (methane in our study).

As shown in Figure 1, the modulated laser beam is focalized above the center of the resonator so that once the central part moves downward by the action of the acoustic wave, the H-square suspended arms are deflected upward and vice versa. Consequently, the gap reduction between H-square suspended arms (movable electrode) and the substrate (fixed electrode) induces the capacitive signal.

The design approach consists in partitioning the resonator functions into a central zone dedicated to photoacoustic energy collection and side zone dedicated to capacitive transduction. Hence, these two functions can be separately optimized.

To enhance the collection of the photoacoustic energy, the structure size should be selected to resonate at a frequency which corresponds to the maximum acoustic pressure generated by the target gas. This maximum pressure is related to the target gas vibrational–transitional relaxation rate by means of the heat production rate, expressed in [10]. The wide central part is devised to overlap with the acoustic wave and thus increase the applied force, as the wider the surface, the higher the force will be. A hole is etched under the central part to avoid the squeeze film effect. A gap of 50 µm is ensured between the hole edges and the resonator central part. Knowing that the acoustic wavelength at the resonance *λ_a_*∼3.4 cm (*λ_a_* = $\frac{c_{s}}{f_{0}}$, where $c_{s}$ is the speed of sound = 343 m/s and $f_{0}$ is the frequency = 10 kHz), this reduced gap together with the resonator thickness (~75 µm) allows the pressure difference between the upper and back sides of the mechanical resonator to increase. Thus, the applied force is increased, as illustrated in [10].

To enhance the resonator displacement, the mechanical susceptibility and the structure quality factor should be maximized. For that, the central back etched hole and the thin separated arms of 12 µm width contribute to prevention of viscous damping. It is important to note that the target resonant mode must be the one where the arms of both sides of the H-square resonator move vertically in-phase (in the same direction). This mode allows to ensure the most efficient conversion of mechanical movement into capacitive signal. In addition, the supports, which holds the structure, are placed at the nodes of the target resonant mode to avoid motion restraints. All these features ensure a high displacement ratio between the extremities of the arms and the central part of the resonator. Consequently, the mechanical energy dissipation and the effective mass, which represents the mass part of the structure involved into motion, are reduced. This increases the mechanical susceptibility and thus, the structure sensitivity, to enhance the signal-to-noise ratio (SNR) [10].

Regarding the structure side parts devoted to capacitive transduction, perpendicular arms together with horizontal ones are employed to simultaneously increase the nominal capacitance, the structure stiffness, and the separation between resonant modes [11]. However, this increases the effective mass, thus, minimizing the displacement of the arms and consequently inducing a reduced capacitive signal. To provide a good compromise, the structure mass is lightened by limiting the number of arms.

In order to predict the design performance, parametric studies were first conducted using the finite element modelling COMSOL software together with an analytical model, presented in [10]. Combining numerical and analytical simulations allows us to determine the optimal design of the MEMS by estimating, for each design, a theoretical detection limit for methane. Figure 2a depicts a simulation that illustrates the movement of the MEMS at its resonance frequency. The arms of the resonator exhibit a greater displacement than its central part, which enables the reduction of effective mass while maintaining a large surface area for collecting acoustic forces. This configuration allows for optimal performance and sensitivity of the MEMS device. Once the numerical/analytical prediction fulfills previously mentioned and required improvements, the mechanical resonator is fabricated using the same process as the H-resonator, detailed in [11].

For the measurements, the substrate is polarized with a DC bias voltage $V_{DC}$. Figure 2b presents the fabricated H-square resonator, with its aluminum wire bonding to measure the output capacitive current.

## 3. Electrical Characterization

The electrical characterization aims to determine the H-square frequency response, quality factor, and nominal capacitance. Figure 3 illustrates the electrical characterization circuit, with an equivalent linear small-signal circuit used to describe the mechanical resonator. A DC polarization voltage ($V_{DC}$) and an AC drive voltage ($V_{AC}$) are applied to one electrode of the mechanical resonator. A transimpedance amplifier (TA) then keeps the second electrode at the electrical ground.

According to Butterworth–Van Dyke model [12], the H-square resonator can be modeled as a series *RLC* circuit with a parallel capacitance $C_{0}$. An electrical *RLC* and a mechanical Mass-Spring-Damper (MSD) system have same form of differential equations. If we consider by analogy that the voltage represents the force, *R* is proportional to the damping, representing the system losses, *L* is proportional to the effective mass, and *C* is inversely proportional to the stiffness. $C_{0}$ accounts for fixed electrical feedthrough capacitance and can be expressed as follows:(1)$$C_{0}=C_{arms}+C_{p}$$

$C_{arms}$ is the capacitance caused by the 3 µm air gap between the suspended H-square arms and the substrate. As presented in Figure 1, the arms are the movable part of the H-square that generate the output electrical signal. By opposition, $C_{p}$ represents the parasitic capacitance induced by all the other parts of the H-square that contributes to $C_{0}$. The silicon square supports holding the structure and separated from the substrate by a layer of 3 µm silicon oxide layer mostly participate in $C_{p}$. The output current, generated by the displacement of the H-square arms, is amplified and converted into voltage using a transimpedance amplifier (TA) with 10^8^ gain, followed by a lock-in amplifier.

In such a damping circuit, the H-square admittance $Y_{Hsq}$ (Ω^−1^) varies as follows:(2)$$Y_{Hsq}=\frac{1}{Z_{Hsq}}=\frac{V_{out}}{\begin{array}{c} G_{TA}\cdot V_{exc} \end{array}}$$
where $Z_{Hsq}$ and $G_{TA}$ are the H-square equivalent linear small-signal circuit impedance (Ω) and *TA* gain (V/A), respectively.

The frequency response of the H-square resonator is analyzed under different $V_{DC}$ bias voltages, as shown in Figure 4. The frequency profile is composed of a first main resonance peak followed by an anti-resonance/opposite peak due to the feedthrough capacitance $C_{0}$. In addition, the effect of $C_{0}$ appears far from the resonance where the H-square resonator is immobile, and thus, where $i_{res}=0$. This leads to an output current $i_{out}=i_{C0}$ and an output voltage $V_{out}$ which can be expressed as follows:(3)$$V_{out}=G_{TA}\cdot i_{C0}\;V_{out}=G_{TA}\cdot V_{exc}\cdot C_{0}\cdot 2\cdot \pi \cdot f$$
where f is the drive potential $V_{AC}$ frequency.

This simply allows $C_{0}$ to be deduced:(4)$$C_{0}=\frac{V_{out}}{G_{TA}\cdot \;V_{exc}\cdot 2\cdot \pi \cdot f\;}$$

All the experimental data are fitted using the Butterworth–Von Dyke model, thus allowing to estimate the resonance frequency $f_{0}$, quality factor Q, and feedthrough capacitance $C_{0}$. Based on the dimension of the H-square design, the theoretical *C*_0_ and *C*_p_ are equal to 5.66 and 3.11 pF, respectively. Compared to the measured $C_{0}$ (=5.51 pF), the relative error is about 3%. As illustrated in Equation (2), to improve the sensitivity of the developed sensor, $C_{0}$ must be maximized without increasing $C_{p}$. The output signal attenuation can be evaluated according to this ratio: $C_{0}/\left(C_{0}+C_{p}\right)$ [5]. Here, the resulting attenuation factor is 0.65, higher than that of the previous H-resonator (0.18) [11].

Figure 4 clearly shows a resonance frequency shift with the polarization voltage. This shift is caused by the DC force generated by the polarization. Moreover, the signal increases with the polarization voltage. However, it is important to know that beyond a certain limit, a further increase of the bias voltage may lead to non-linearities and the collapse of the resonator, as observed in [13]. The capacitive signal generated by the deflection of H-square resonator is maximum with a polarization voltage of 17.5 V, inducing a resonance frequency of ~10.3 kHz and Q of ~22. This exact same polarization voltage will be further applied for the following measurements of this study.

## 4. Photoacoustic Sensing

### 4.1. Photoacoustic Signal

It is important to highlight that the H-square resonance frequency corresponds to an acoustic pressure close to the maximum of pressure generated through the absorption of the modulated laser light by methane molecules. This maximum pressure is correlated to the molecular relaxation time of methane ~11.5 μs via the heat production rate [10,14]. Figure 5 illustrates the experimental setup suited to MEMPAS. It mainly consists of a frequency modulated laser source, a collimating lens with high numerical aperture, and an aluminum gas chamber in which the H-square resonator is placed, transimpedance amplifier and lock-in amplifier for signal processing. The gas cell has two outlets for gas circulation and is equipped with CaF_2_ sealed windows tilted at 20 degrees to prevent interferences and feedback into the laser. The measurements are performed under room temperature.

As shown in Figure 1, the modulated laser beam is precisely positioned perpendicular and focalized above the center of the resonator so that the acoustic wave mainly overlaps with the resonator central zone. Indeed, this perpendicular configuration reduces the overlapping of the acoustic wave with the H-square arms, which are in phase opposition with the central part. The action of the acoustic wave deflects the H-square central part, and thus, the mechanically coupled suspended arms. Therefore, the capacitive signal is induced thanks to the gap reduction between H-square suspended arms and the substrate.

Based on the HITRAN database [15], among the multiple absorption peaks of methane, the target line to detect is a strong and sufficiently separated peak located at ν = 4300.36 cm^−1^ and characterized by absorption cross-section σ = 8.52 · 10^−21^ cm^2^. Molecules^−1^. For that, a modulated NORCADA distributed feedback (DFB) laser emitting around 2.3 µm (~4300 cm^−1^) with an output power of 3.9 mW is used. The chosen injected current is 140 mA at 25 °C.

The cell is filled up with 1% CH_4_ diluted in pure nitrogen. A spectral scan was realized by tuning the laser current between 136 mA and 145 mA. A preliminary direct absorption test is performed using a photodiode placed on the optical axis on the backside of the gas cell (TDLS configuration). Then, to demonstrate the feasibility of newly developed H-square mechanical resonator, the first and second harmonic detections (1f and 2f) are carried out in order to prove that the signal comes from photoacoustic and not from an eventual photothermal excitation. The laser is modulated at the resonance frequency of the H-square $f_{0}$/n, which is ~10.28 kHz for the first harmonics (n = 1) and at ~5.14 kHz for the second harmonics (n = 2). The laser optimum amplitude of modulation is reached at 9.6 mA and 7.2 mA for 1f and 2f detection, respectively. This allows maximizing the generated acoustic pressure and, thus, the H-square arms displacement, to get the highest signal-to-noise ratio.

Figure 6 shows the 1f and 2f signatures delivered from the H-square resonator compared to the gas absorption spectrum measured with a photodiode. The 1f and 2f correspond to the first and the second derivative of the absorption peak of methane, respectively. These results allow determination of the laser currents for which the 1f and 2f signals reach their maxima.

Table 1 shows the optimum laser modulation parameters to obtain the 1f and 2f maxima, the resulted laser power, and the corresponding effective power. This latter corresponds to the proportion of the laser power used to generate the acoustic wave at the desired frequency [10]. These operating conditions are used for the following sections of this paper.

To estimate the displacement of H-square electrodes (arms) under photoacoustic excitation, a Polytec OVF-5000 laser Doppler Vibrometer (LDV) is used. The 1f detection is performed on the focal point of the LDV laser spot (3 µm diameter) at the center and the left and right extremities of the H-square resonator. By applying a frequency sweep, the results, shown in Figure 7, reveal that the H-square arms amplitude of displacement reaches 20 pm (picometers) for 1% methane concentration. The difference between the displacement of the left and right arms of the H-square resonator may be due to a fabrication defect. As predicted, the displacement amplitude of the central part is much lower than that of the suspended arms. The simulated displacement ratio between the extremities of the arms and the central part is 12.8, showing a good agreement with the measured displacement ratio of 14.8 for the left arm and 8.3 for the right arm. These displacement ratios are three times higher than that of the previous H-resonator [11].

It is important to note that the reduced displacement of the central part contributes to lowering the structure’s effective mass. On the other hand, this contributes to increasing the displacement of the H-square arms, therefore increasing the output signal.

### 4.2. Capacitive Calibration

The linearity of the sensor was evaluated by monitoring the 1f photoacoustic signal for different methane concentrations. These latter are obtained using a mass flow controller, Alytech GasMix Aiolos II. The calibration process consists in measuring continuously (with an integration time of 1 s) the photoacoustic signal, while successively filling the cell with a calibrated methane concentration and flushing with N_2_ between each concentration level to check the recovery of the baseline. Each step is conducted for 10 minutes duration with a gas flow rate of 950 NmL/min. Figure 8 shows the recorded measurement of the H-square signal as a function of the injected methane concentrations. A reproducible output signal baseline reappears after flushing with N_2_ which confirms the consistency of the resonator response. The results prove the linearity of the sensor response and make the sensor easy to calibrate for real-life applications.

### 4.3. Allan–Werle Deviation

To assess the sensor performance, the Allan–Werle deviation has been performed, using the AllanTools Python library. It presents an excellent indicator of the sensor stability and limit of detection (LOD), i.e., lowest detectable concentration.

For any photoacoustic gas sensor, the signal-to-noise ratio (SNR) depends on the laser power used, the effective absorption cross section of the gas, and the detection bandwidth. To characterize the sensitivity of a gas sensor, it is common to use the NNEA or Normalized Noise Equivalent Absorption, which is a quantity normalized with of these three factors. By definition, it is the minimum detectable absorption using 1 W of power and a detection bandwidth of 1 Hz. Thus, the lowest value of NNEA gives the highest performances. It can be expressed as follows:(5)$$\mathrm{NNEA}=\frac{\;\alpha_{\mathrm{LOD}}\cdot \;P}{\sqrt{\Delta f}}$$
where $\alpha_{\mathrm{LOD}}$ is the absorption coefficient at the limit of detection, P is the laser power, and Δf is the integration bandwidth.

The experiment consists in recording the 1f and 2f signals for 60 min with a CH_4_ concentration of 1% and a time constant of 100 ms. The applied laser modulation parameters are displayed in Table 1. For a given integration time, the Allan–Werle deviation is calculated, as shown in Figure 9. The deviation exhibits a similar tendency for both 1f and 2f modes. For an integration time τ < 1 s, a drop-off appears due to the cut-off frequency of the lock-in amplifier low pass filter. The Allan deviation is trustworthy for an integration time τ that is at least five times greater than the lock-in amplifier time constant [16]. As shown in Figure 9, for τ > 500 ms, the curves have the classical τ^−1/2^ slope indicating dominant white noise. The shape of this curve reflects the stability of the sensor response. Long-term drift appears after 400 s and 700 s for 1f and 2f modes, respectively. The LOD is 104 ppmv for 1 s of integration time and 38 ppmv for 10 s in the 1f mode. For 2f mode, the LOD accounts for 175 ppmv for 1 s of integration and 65 ppmv for 10 s of integration. The LOD reaches 7 ppmv for both 1f and 2f modes before their corresponding drifts.

The analysis of the Allan deviation for 1f detection at 1 s integration time exhibits an LOD leading to a normalized noise equivalent absorption coefficient (NNEA) of 8.6 · 10^−8^ Wcm^−1^ Hz^−1/2^. This makes the performance of the H-square resonator superior to that of a bare QEPAS-based sensor characterized by a NNEA = 1.3 · 10^−7^ Wcm^−1^ Hz^−1/2^ and that of the previous H-resonator with a NNEA = 5.5 · 10^−7^ Wcm^−1^ Hz^−1/2^, both measured using the same experimental setup as the H-square resonator [11].

## 5. Discussion and Perspectives

The work presented in this paper was motivated by the need for a highly sensitive, selective, and compact gas sensor. To meet this objective, a new photoacoustic technique, dubbed MEMPAS, is developed. This method consists of using a resonant capacitive MEMS specifically designed for photoacoustic gas sensing with higher quality factor compared to microphones. The proposed H-square resonator represents an improved generation of an early stage developed resonator characterized by its ability to limit viscous damping. It is composed of a wide central part dedicated to the collection of photoacoustic energy and side-parts of thin arms dedicated to the capacitive transduction. Unlike the QTF, the design maximizes the overlap of the acoustic wave through the surface of the central part while maintaining a reduced effective mass. This is ensured thanks to the high displacement ratio between the side-parts and the central part of the H-square resonator.

In this work, the photoacoustic energy collection is improved by increasing the pressure difference between the upper and back sides of the mechanical resonator. This is performed by setting a reduced gap of 50 µm between the resonator central part and the edges of the hole, which have been etched underneath. Additionally, the resonator length is chosen to operate at a frequency which corresponds to an acoustic pressure close to the maximum of pressure generated due to the absorption of the laser light by the target gas. Furthermore, to reduce the support dissipation, the supports holding the structure are placed at the nodes. Regarding the improvements for the capacitive transduction, perpendicular and horizontal thin separated arms were employed to simultaneously increase the nominal capacitance, the structure stiffness, and the separation between resonant modes.

In this article, we have demonstrated the operating principle of our gas sensor using methane. However, it is worth noting that this same sensor can be used to detect a wide range of gases, not just methane. This is due to the fact that our gas sensor is based on photoacoustic spectroscopy. By replacing the laser used in the sensor, it can detect any gas, making it highly versatile and adaptable to various applications.

The electrical characterization confirms that the H-square resonator operates at a frequency of 10.28 kHz, which corresponds to acoustic pressure close to the maximum of pressure generated due to the absorption of the laser light by the target gas. It also reveals that the nominal capacitance and output signal attenuation are increased in comparison with the previous H-resonator [11]. The photoacoustic experiment demonstrates the viability of the H-square resonator for methane trace detection. The linear response of the resonator is validated on calibrated methane concentrations.

Concerning the quality factor, the H-square exhibits a quality factor of ~22, which is low compared to the QTF quality factor, estimated to be ~10,000. It is important to note that under vacuum, piezoelectric excitation of the H-square leads to a quality factor of 1570. This reflects the significant effect of the viscous damping on the total quality factor of the H-square due to its low resonance frequency. Further design optimizations are needed to achieve higher quality factor, mainly by improving the viscous and the support quality factor [10].

In terms of sensitivity, the H-square resonator performances are superior by a factor of 6.4 compared to the previous H-resonator and by a factor of 1.5 compared to the bare QEPAS, a state-of-the-art reference to compact and selective gas sensors. Hence, by optimizing the MEMS design and maximizing its sensitivity, the objective of this research is to obtain the best possible sensor performance. With this accomplishment, the next step would be to integrate an acoustic cavity to further enhance the sensor sensitivity and signal-to-noise ratio (SNR). For instance, in QEPAS, the acoustic cavity allows to yield a factor of 30 [17,18,19]. In our case, the gain is expected to be even higher than in Cantilever-Enhanced Photoacoustic Spectroscopy (CEPAS), where the coupling between the acoustic cavity and the cantilever is optimized. The possibility of achieving such perspective allows obtaining a highly performant gas sensor among photoacoustic techniques while remaining compact.

## Figures and Tables

**Figure 1 sensors-23-03280-f001:**
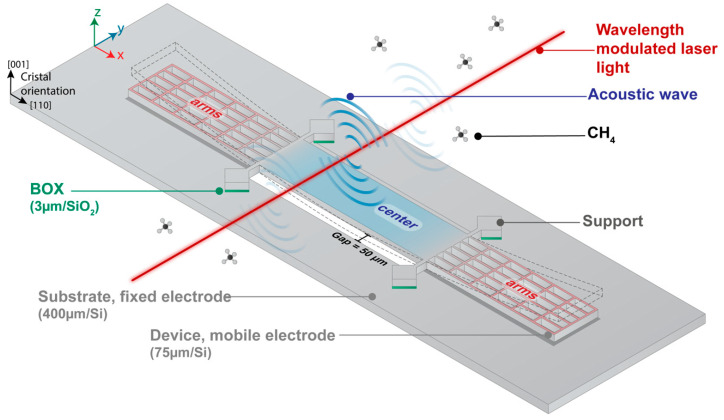
Schematic illustration of H-square resonator deflection under photoacoustic excitation. The resonator is divided in two parts: the center part dedicated to the photoacoustic energy collection and the arms dedicated to the capacitive transduction. The optical axis of the laser beam is set perpendicular to the resonator and focalized on its middle. The acoustic wave is generated due to the absorption of modulated laser light.

**Figure 2 sensors-23-03280-f002:**
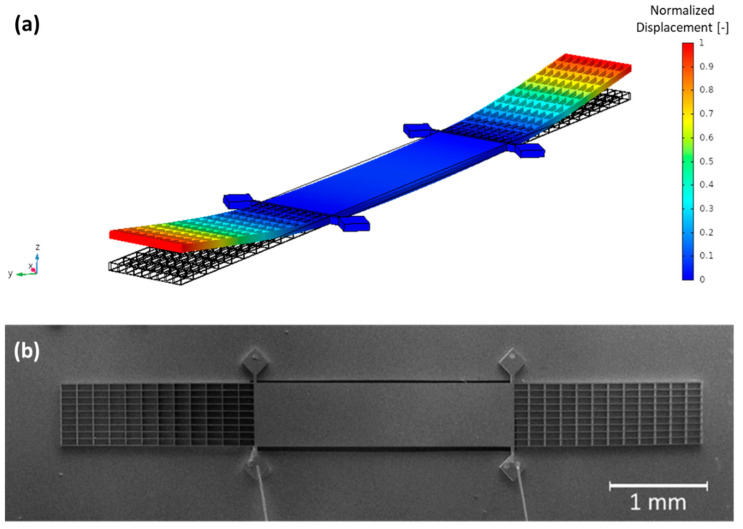
(**a**) Simulation results of first mode of vibration of H-square resonator obtained by using COMSOL Multiphysics. (**b**) Scanning electron microscopy image of the silicon-based micro-mechanical resonator (H-square resonator) with its aluminum wire bonding. The resonator dimensions are the following: length 8530 μm, width 805 μm, and thickness 75 μm.

**Figure 3 sensors-23-03280-f003:**
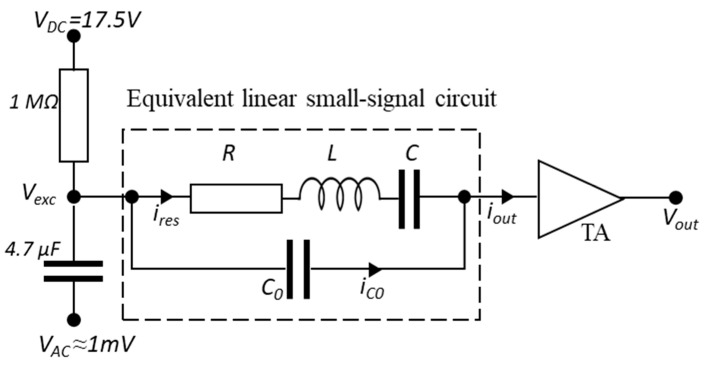
Electrical characterization circuit. $V_{DC}$, $V_{AC}$, $V_{exc}$_,_ and $V_{out}$ are the polarization voltage, the drive voltage, excitation voltage, and the output voltage, respectively. The output current $i_{out}$ is composed of $i_{res}$ and $i_{C0}$, which are the currents passing through the *RLC* and $C_{0}$ branch.

**Figure 4 sensors-23-03280-f004:**
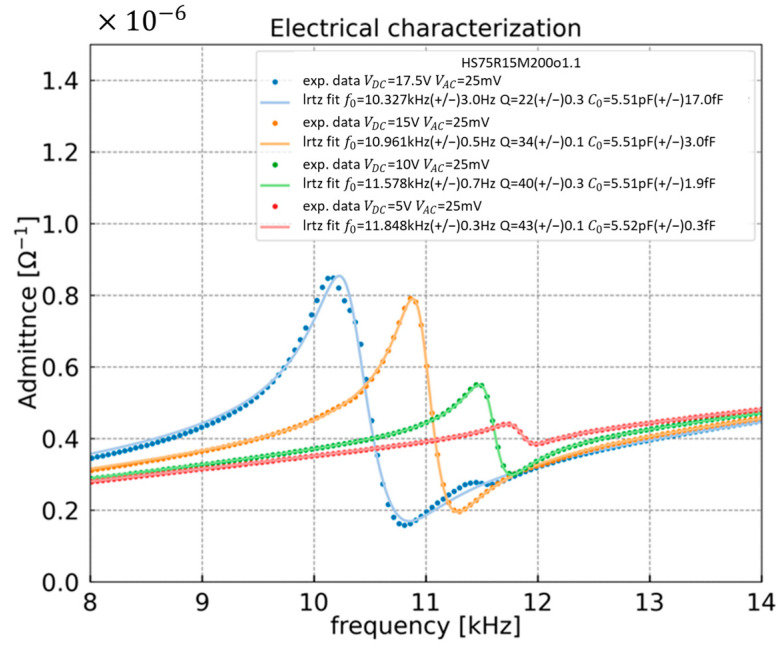
The electrical admittance variation as function of the frequency for a 25 mV drive voltage and for different polarization voltages of 5, 10, 15, and 17.5 V. The lines represent the fitted curves while points are the experimental data.

**Figure 5 sensors-23-03280-f005:**
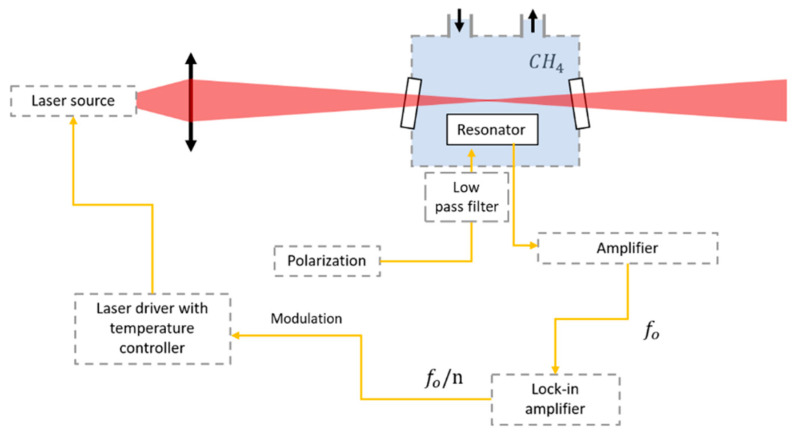
Schematic of the MEMPAS setup for methane detection [11].

**Figure 6 sensors-23-03280-f006:**
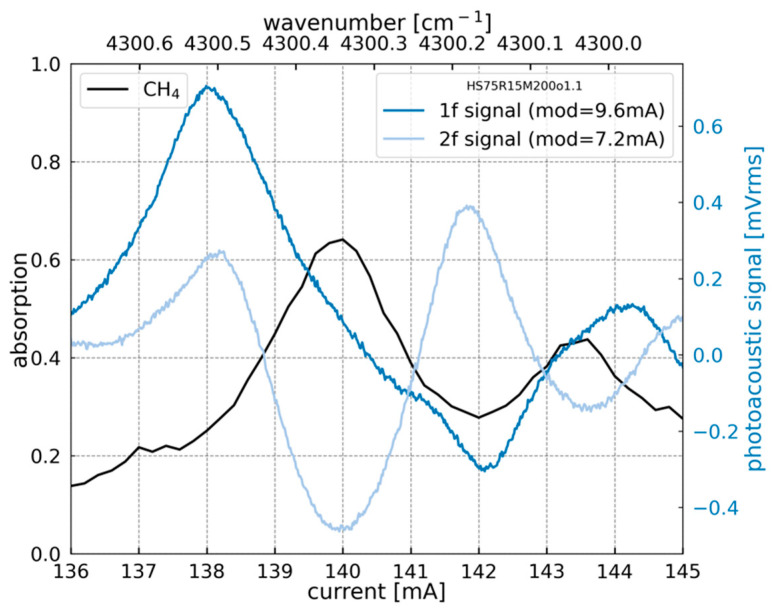
Absorption spectrum of methane (black curve) measured using a photodiode and the photoacoustic signal detected via the capacitive response of H-square resonator in 1f (dark-blue) and 2f (light-blue) modes at 1% of CH_4_ at atmospheric pressure using a DFB laser emitting around 2.3 μm with an output power of 3.9 mW (at 140 mA).

**Figure 7 sensors-23-03280-f007:**
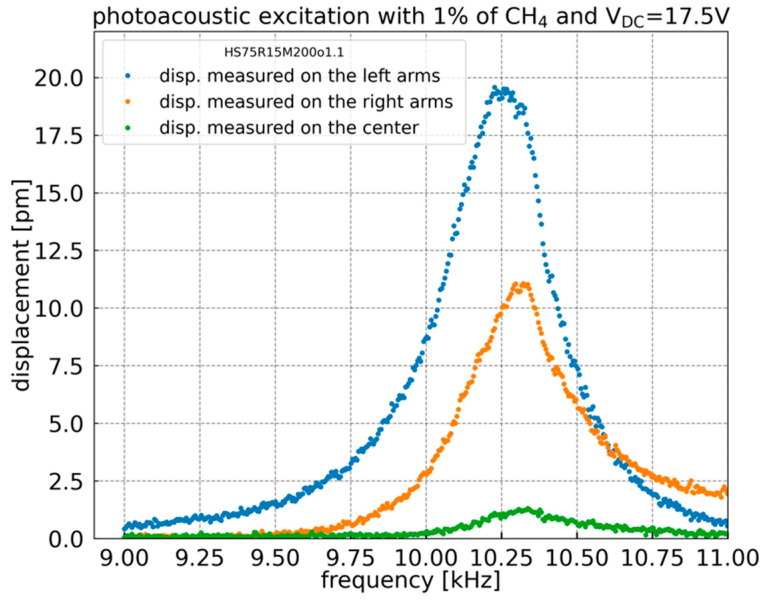
Displacement extracted from LDV measurement at the center and the left and right extremities of the H-square resonator at 1% of CH_4_. Polytec OVF-5000 LDV using VD-06 decoder, 2 mm/s/V range.

**Figure 8 sensors-23-03280-f008:**
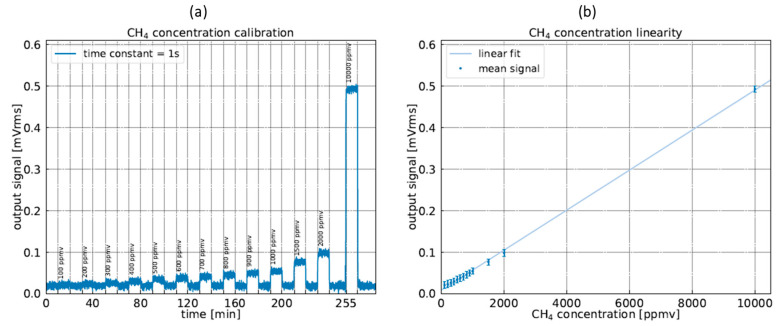
(**a**) H-square signal at the first harmonic (1f) during a gas step cycle, with concentration ranging from 100 ppmv to 2000 ppmv and at 10,000 ppmv. The calibrated dilution of methane is injected for 10 min, then the cell is flushed with pure N_2_ for 10 min. The integration time is set to 1 s. (**b**) H-resonator signals as a function of the injected methane concentration. The calibration curve is obtained by applying a linear fit.

**Figure 9 sensors-23-03280-f009:**
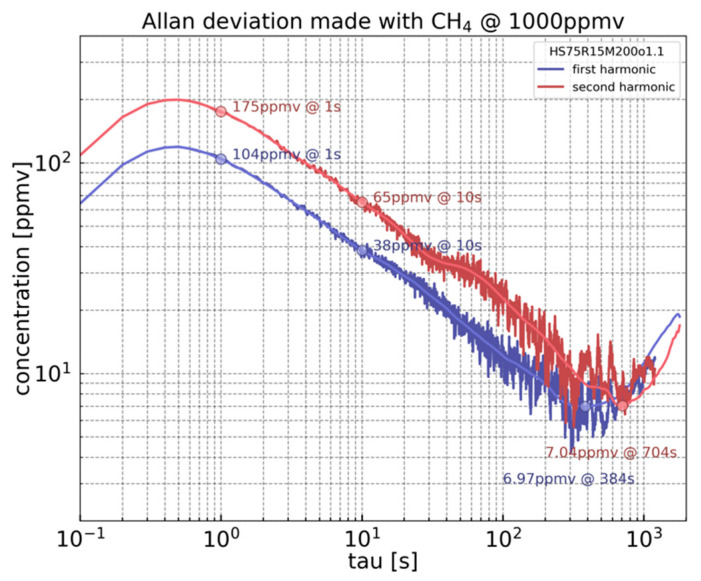
Allan–Werle deviation calculated from a 60 min acquisition for the 1f and 2f mode with 1% of CH_4_ concentration and time constant of 100 ms.

**Table 1 sensors-23-03280-t001:** Laser modulation parameters to monitor the maxima of 1f and 2f signals.

Laser Modulation Parameters	1f Signal	2f Signal
Fixed current (mA)	138	140
Modulation amplitude (mA)	9.6	7.2
Modulation frequency (kHz)	10.28	5.14
Resulted Laser power (mW)	3.86	3.92
Effective power (mW) [10]	1.93	1.37

## Data Availability

The data presented in this study are available on request from the corresponding author.

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
