# Peer review of "Highly Sensitive Capacitive MEMS for Photoacoustic Gas Trace Detection"

_sensors, 2023, doi:10.3390/s23063280_

Round 1
Reviewer 1 Report
The paper presents a study of a MEMS detector for gases. The presented results are well described and the conclusions are well supported by the data provided, supported by the Allan curve.
In the conclusion, the authors compare the obtained measurements with other photoacoustic detectors. Although comparing with other gas detection techniques principles is quite difficult out of pure ppm detection performances, some numbers either in the introduction or conclusion could have been useful.
Author Response
Answer: It is true that the detection performance of a sensor to detect gas depends on various factors, such as the sensor technology, design, and operating conditions. This is why in this study, under the same operating conditions, we compare the performance of our sensor with bare Quartz-Enhanced Photoacoustic Spectroscopy (QEPAS), a state-of-the–art reference to compact and selective gas sensors.
However, we decided to add line 26 a brief comparison of the most commonly used gas sensor: “Electrochemical and semiconductor sensors are characterized by their compactness and relatively low cost. Electrochemical sensors detect electron transfer events during electrochemical reactions, while semiconductor sensors operate by measuring changes in electrical resistance caused by the adsorption of target gas molecules. Both types of sensors are sensitive to the detection of gases like methane where the limit of detection can reach less than 9 ppmv and 5 ppmv for electrochemical and semiconductors sensors, respectively [2,3]. However, these sensors have decreased reliability and limited lifetime due to the impact of humidity and temperature, as well as the gradual consumption associated with chemical reactions. In addition, cross-sensitivity can occur for some gases, making these gas sensors less selective.”
We added also when introducing TDLS line 36: “, capable of reaching ppb levels, as illustrated in [4] for methane detection.”
The added references are in lines 412-418:
“ [2] M. Dosi, I. Lau, Y. Zhuang, D.S.A. Simakov, M.W. Fowler, M.A. Pope, Ultrasensitive Electrochemical Methane Sensors Based on Solid Polymer Electrolyte-Infused Laser-Induced Graphene, ACS Appl. Mater. Interfaces. 11 (2019) 6166–6173. https://doi.org/10.1021/acsami.8b22310.
[3] Z.P. Tshabalala, K. Shingange, B.P. Dhonge, O.M. Ntwaeaborwa, G.H. Mhlongo, D.E. Motaung, Fabrication of ultra-high sensitive and selective CH4 room temperature gas sensing of TiO2 nanorods: Detailed study on the annealing temperature, Sens. Actuators B Chem. 238 (2017) 402–419. https://doi.org/10.1016/j.snb.2016.07.023.
[4] M. KwaÅ›ny, A. Bombalska, Optical Methods of Methane Detection, Sensors. 23 (2023) 2834. https://doi.org/10.3390/s23052834.”

Reviewer 2 Report
Please find my comments in the attached file.

Author Response
- Line 22: “and” in “of and toxic” seems not to be needed.
Answer: Thank you. We removed ‘and’ from line 22.
- Lines 44-47: semicolons and a dot are suggested to be added at the ends of the
lines depicting PA techniques.
Answer: Thank you. We added the suggested semicolons and dots at the ends of Lines 51-55.
- Line 46: there are too many spaces before “forks”.
Answer: Thank you. We remove spaces before “forks” from line 54. In addition, we added an abbreviation ‘(QTF)’.
- Line 62: “accelerometers” should be more precisely referred to those using
capacitive sensing as the sentence sounds a little bit out of topic.
Answer: To avoid potential confusion, we decided to remove “For instance, in accelerometers, the accuracy can reach few ppm” from line 70.
- Line 70: in my opinion, ‘photoacoustic energy’ or ‘photoacoustic pressure’ seems
to be better than “photoacoustic force” (the same applies to line 75 and the
following). The PA detection relies on catching energy of acoustic waves, i.e.
periodical pressure variations, and there is no such physical quantity as
‘photoacoustic force’ - the variable force is exerted by the gas pressure variations
related to PA absorption.
Answer: Thank you for the suggestion. We decided to replace ‘photoacoustic force’ by ‘photoacoustic energy’in lines 78,83 and 358. In addition, we replaced ‘photoacoustic/acoustic force’ by the ‘applied force’ in lines 123,124 and 128.
- Lines 76-77: it should be clearly stated that the objective of the study is to develop
a methane PA sensor not a ‘universal’ PA sensor for gas trace detection (as the
reader can now imagine).
Answer: Indeed, maybe this part is not clear enough, our sensor can be used to detect a wide range of gases, not just methane. Gas sensor based on photoacoustic detection, offer the possibility to address a wide variety of gas, by simply replacing the exciting laser source. We decided to add line 367 “In this article, we have demonstrated the operating principle of our gas sensor using methane. However, it is worth noting that this same sensor can be used to detect a wide range of gases, not just methane. This is due to the fact that our gas sensor is based on photoacoustic spectroscopy. By replacing the laser used in the sensor, it can detect any gas, making it highly versatile and adaptable to various applications.”
- Line 78: It should be explained why authors avoid using an acoustic cavity in their
PA MEMS sensor design.
Answer: This work focuses on the development of an acoustic sensor based on the MEMS technology. The primary objective of this research is to achieve the best possible sensor performance. By optimizing the sensor design and maximizing its sensitivity and signal-to-noise ratio, this work aims to improve the accuracy and precision of the sensor measurements. Once this has been achieved, the sensor can be integrated with an acoustic cavity to further enhance its performance and sensitivity.
To provide further clarification, we add line 384 “In terms of sensitivity, the H-square resonator performances are superior by a factor of 6.4 compared to the previous H-resonator and by a factor of 1.5 compared to the bare QEPAS, a state-of-the-art reference to compact and selective gas sensors. Hence, by optimizing the MEMS design and maximizing its sensitivity, the objective of this research is to obtain the best possible sensor performance. With this accomplishment, the next step would be to integrate an acoustic cavity to further enhance the sensor sensitivity and signal-to-noise ratio (SNR).For instance,” and we remove from line 388:“Although the H-square reaches higher performances than the bare QEPAS, the use of an acoustic cavity is required to enhance the acoustic force and prevent any potential photothermal noise from reducing the signal-to-noise ratio (SNR).”
- Line 80: although LOD is a quite well-known quantity yet normalised noise
equivalent absorption (NNEA) is a highly specialized term. It’s meaning is
mentioned then in lines 295-296; however, it seems reasonable to explain in more
detail the meaning and importance of this term to a reader who may not be an
expert in PA or signal processing.
Answer: Thank you for pointing that out. We decided to provide more explanation on the NNEA line 319 “For any photoacoustic gas sensor, the signal-to-noise ratio (SNR) depends on the laser power used, the effective absorption cross section of the gas and the detection bandwidth. To characterize the sensitivity of a gas sensor, it is common to use the NNEA or Normalized Noise Equivalent Absorption, which is a quantity normalized with of these three factors. By definition, it is the minimum detectable absorption using 1 W of power and a detection bandwidth of 1 Hz. Thus, the lowest value of NNEA gives the highest performances. It can be expressed as follows:
|
|
(1) |
where αLOD is the absorption coefficient at the limit of detection, P is the laser power and Δf is the integration bandwidth.”
- Line 85: “the device” should be written in capital T.
Answer: Thank you. We replaced it by capital “T” in line 94
- Line 86: the resistivity (actually the volume resistivity) is given in “Ω·cm” not in “Ω.
cm−1” as stated in the document. Please correct.
Answer: Thank you. We replaced it by “Ω·cm” line 95
- Figure 1: ‘Substrat’ should be corrected to “Substrate”.
Answer: Thank you. We corrected to ‘substrate’ in Figure 1
- Line 111: “acoustic force” – please see my remark in p. 5.
Answer: Thank you. Please see the answer to P.5
- Line 113: It should be explained how the resonance acoustic wavelength was
calculated.
Answer: we removed “λa∼ 3.5 cm” and added line 126 ‘the acoustic wavelength at the resonance λa ∼ 3.4 cm (λa =cs/f, where cs is the speed of sound=343m/s and f is the frequency=10kHz)’.
- Lines 134-135: A graphical illustration of the FEM simulation would provide a
valuable insight into the design and the resonant operation of the presented
MEMS device. It should also be used to explain the dimensioning of the device,
since currently all the dimensions given in the text have no direct link to the
methane PA resonant sensing.
Answer: We appreciate your suggestion. we added a graphical illustration of the FEM simulation (Figure 2.a). In addition, we added line 149: “Combining numerical and analytical simulations allows us to determine the optimal design of the MEMS by estimating, for each design, a theoretical detection limit for methane. Figure 2.a. depicts a simulation that illustrates the movement of the MEMS at its resonance frequency. The arms of the resonator exhibit a greater displacement than its central part, which enables the reduction of effective mass while maintaining a large surface area for collecting acoustic forces. This configuration allows for optimal performance and sensitivity of the MEMS device.”
- Line 153: a literature reference should be provided for the Butterworth-Van Dyke
model of H-square resonator operation.
Answer: we added literature reference “[12]” in line 174 and reference line 436: “[12] J.-M. Friedt, É. Carry, Introduction to the quartz tuning fork, Am. J. Phys. 75 (2007) 415–422. https://doi.org/10.1119/1.2711826.”
- Equations (2), (3) and (4): dots used as multiplication symbols should be located
at the half-height of the symbols (not at their baseline); please correct.
Answer: Agree. we corrected the dots positions in lines 198, 207 and 210 corresponding to Equations (2), (3) and (4).
- Figure 4: the axis legend should not contain the admittance definition as it was
already defined earlier in the text (in (2)).
Answer: Agree. we remove the admittance definition from the graph (Line 211).
- Line 206: It should be clearly explained why the H-square resonant frequency of
10.3 kHz was selected as the optimal and how it is related to the information
presented in line 211, related to the molecular relaxation time of methane given
as close to 11.5 μs. The inverse of the provided relaxation time gives a frequency
of 86,9 kHz, which is far from the selected 10.3 kHz resonance (and it is not its
overtone as 86,9/10,3≈8,4). Please explain or correct.
Answer: In the text, it was not explained in detail. Indeed, the maximum acoustic pressure cannot be simply derived from the relaxation time. Our explanation referred to article: ”[10] Trzpil, N. Maurin, R. Rousseau, D. Ayache, A. Vicet, M. Bahriz, Analytic Optimization of Cantilevers for Photoacoustic Gas Sensor with Capacitive Transduction, Sensors. 21 (2021) 1489. https://doi.org/10.3390/s21041489.”
To clarify this, we reformulated and replaced:
- in line 120 “To enhance the collection of the photoacoustic energy, the structure size should be selected to resonate at a frequency which corresponds to the target gas optimum vibrational-transitional relaxation rate” by “To enhance the collection of the photoacoustic energy, the structure size should be selected to resonate at a frequency which corresponds to the maximum acoustic pressure generated by the target gas. This maximum pressure is related to the target gas vibrational-transitional relaxation rate by means of the heat production rate, expressed in [10].”
- in line 232 “It is important to highlight that the H-square resonance frequency is optimized according to the molecular relaxation time of methane (CH4) ~ 11.5 μs [12].” by “It is important to highlight that the H-square resonance frequency corresponds to an acoustic pressure close to the maximum of pressure generated through the absorption of the modulated laser light by methane molecules. This maximum pressure is correlated to the molecular relaxation time of methane ~ 11.5 μs via the heat production rate [10,15].”
- in line 360 “Additionally, the resonator length is chosen to operate at a frequency, which corresponds to the target gas vibrational-transitional relaxation time.” by “Additionally, the resonator length is chosen to operate at a frequency, which corresponds to an acoustic pressure close to the maximum of pressure generated due to the absorption of the laser light by the target gas.”
- in line 372 “The electrical characterization confirms that the H-square resonator operate at a frequency which corresponds to the molecular relaxation rate of methane, which is 11 kHz.” by “The electrical characterization confirms that the H-square resonator operate at a frequency of 10.28 kHz which corresponds to an acoustic pressure close to the maximum of pressure generated due to the absorption of the laser light by the target gas.”
- Table 1: the table caption should be positioned over the table (not below it). Answer: we corrected the position of the table caption to be over the table (line 281)
- Line 305: ‘7 ppm’ - ppmv should be used consistently throughout the text. Answer: we corrected and replaced “ppm” by “ppmv” in line 337
- Line 327: “photoacoustic force” – please see my remark in p. 5.
Answer: Thank you it has been corrected, please see the answer to P.5
- Line 336: “molecular relaxation rate of methane, which is 11 kHz” - In line 211 the
molecular relaxation time of methane was given to be close to 11.5 μs so there is
a huge discrepancy in those numbers. Correct.
Answer: Thank you it has been corrected, please see the answer to P.18
- It would be valuable to compare the LOD of the developed methane MEMS PA
sensor to other types of methane sensors (both laboratory designs and
commercial ones).
Answer: Thank you it has been corrected. Reviewer 1 made the same comment.
Indeed, the detection performance of a sensor to detect gas depends on various factors, such as the sensor technology, design, and operating conditions. This is why in this study, under the same operating conditions, we compare the performance of our sensor with bare Quartz-Enhanced Photoacoustic Spectroscopy (QEPAS), a state-of-the–art reference to compact and selective gas sensors.
We decided to add line 26 a brief comparison of the most commonly used gas sensor: “Electrochemical and semiconductor sensors are characterized by their compactness and relatively low cost. Electrochemical sensors detect electron transfer events during electrochemical reactions, while semiconductor sensors operate by measuring changes in electrical resistance caused by the adsorption of target gas molecules. Both types of sensors are sensitive to the detection of gases like methane where the limit of detection can reach less than 9 ppmv and 5 ppmv for electrochemical and semiconductors sensors, respectively [2,3]. However, these sensors have decreased reliability and limited lifetime due to the impact of humidity and temperature, as well as the gradual consumption associated with chemical reactions. In addition, cross-sensitivity can occur for some gases, making these gas sensors less selective.”
We added also when introducing TDLS line 36: “, capable of reaching ppb levels, as illustrated in [4] for methane detection.”
The added references are in lines 412-418:
“ [2] M. Dosi, I. Lau, Y. Zhuang, D.S.A. Simakov, M.W. Fowler, M.A. Pope, Ultrasensitive Electrochemical Methane Sensors Based on Solid Polymer Electrolyte-Infused Laser-Induced Graphene, ACS Appl. Mater. Interfaces. 11 (2019) 6166–6173. https://doi.org/10.1021/acsami.8b22310.
[3] Z.P. Tshabalala, K. Shingange, B.P. Dhonge, O.M. Ntwaeaborwa, G.H. Mhlongo, D.E. Motaung, Fabrication of ultra-high sensitive and selective CH4 room temperature gas sensing of TiO2 nanorods: Detailed study on the annealing temperature, Sens. Actuators B Chem. 238 (2017) 402–419. https://doi.org/10.1016/j.snb.2016.07.023.
[4] M. KwaÅ›ny, A. Bombalska, Optical Methods of Methane Detection, Sensors. 23 (2023) 2834. https://doi.org/10.3390/s23052834.”

Reviewer 3 Report
In the manuscript entitled ”Highly sensitive capacitive MEMS for photoacoustic gas trace detection”, an enhanced MEMS capacitive sensor is developed for photoacoustic (PA) gas detection which combines the advantages of silicon technology used in MEMS microphones, and the high-quality factor, characteristic of quartz tuning fork. In the first harmonic detection, the limit of detection (LOD) is 104 ppmv higher than that of bare Quartz-Enhanced Photoacoustic Spectroscopy (QEPAS). This work is interesting and is suggested to be published after minor revision on text editing.
Author Response
Thank you for taking the time to review our work and for your kind words. We are grateful for your feedback and appreciate your recognition of our efforts.
Reviewer 4 Report
The authors presented an original design of a methane sensor based on the photoacoustic principle and characterize its gas sensitive properties. The work is of considerable interest from a scientific point of view and seems promising, but there are several questions.
1. For such thoughtful work, it would be necessary to discuss the process of selectivity and cross-sensitivity primarily to water vapor, that is, to carry out measurements to methane in moist air. Could it be that moisture will be adsorbed on the surface of the silicon resonator, which will lead to a shift in its resonant frequency, for example, and this will degrade the performance of the sensor?
2. At what temperature were the measurements taken? Does temperature affect measurement results?
3. You first write that the relaxation time of the methane molecule is 11.5 microseconds, then that the resonant frequency is 10.28 kHz. How are these quantities related?
4. Line 262. "displacement reaches 20 pm". Can you clarify what this value of "pm" is?
5. Figure 6. How can you explain that the maximum of the photoacoustic signal (dark blue and light blue lines) does not coincide with the absorption maximum of methane (black line)?
6. There are some typos, for example line 281, "950Nml/min", should there be a space instead of "N"?
Author Response
The authors presented an original design of a methane sensor based on the photoacoustic principle and characterize its gas sensitive properties. The work is of considerable interest from a scientific point of view and seems promising, but there are several questions.
For such thoughtful work, it would be necessary to discuss the process of selectivity and cross-sensitivity primarily to water vapor, that is, to carry out measurements to methane in moist air. Could it be that moisture will be adsorbed on the surface of the silicon resonator, which will lead to a shift in its resonant frequency, for example, and this will degrade the performance of the sensor?
Answer: The effects of air humidity on the sensor have not been studied, however, it is expected that the sensor's performance may be affected by air humidity, as referred in [R1] where Rousseau et al. proposed a method to correct signal drift induced by humidity and/or temperature for QEPAS technique. The method consists in implementing QTF characterization in a feedback loop in order to ensure the resonance tracking of the QTF. This Resonance tracking method showed a consistent reduction of the signal drift, less than 1% relative error compared to 44% for conventional QEPAS. This same method can be applied to our MEMS.
[R1] R. Rousseau, N. Maurin, W. Trzpil, M. Bahriz, A. Vicet, Quartz Tuning Fork Resonance Tracking and application in Quartz Enhanced Photoacoustics Spectroscopy, Sensors. 19 (2019) 5565. https://doi.org/10.3390/s19245565.
2. At what temperature were the measurements taken? Does temperature affect measurement results?
Answer: We added line 239 “The measurements were performed under room temperature.”
Yes, the temperature may affect the performances. Please see the answer to Q.1
3. You first write that the relaxation time of the methane molecule is 11.5 microseconds, then that the resonant frequency is 10.28 kHz. How are these quantities related?
Answer: Thank you it has been corrected. Reviewer 2 made the same comment.
In the text, it was not explained in detail. Indeed, the maximum acoustic pressure cannot be simply derived from the relaxation time. Our explanation referred to article ”[10] Trzpil, N. Maurin, R. Rousseau, D. Ayache, A. Vicet, M. Bahriz, Analytic Optimization of Cantilevers for Photoacoustic Gas Sensor with Capacitive Transduction, Sensors. 21 (2021) 1489. https://doi.org/10.3390/s21041489.”
To clarify this, we reformulated and replaced:
- in line 120 “To enhance the collection of the photoacoustic energy, the structure size should be selected to resonate at a frequency which corresponds to the target gas optimum vibrational-transitional relaxation rate” by “To enhance the collection of the photoacoustic energy, the structure size should be selected to resonate at a frequency which corresponds to the maximum acoustic pressure generated by the target gas. This maximum pressure is related to the target gas vibrational-transitional relaxation rate by means of the heat production rate, expressed in [10].”
- in line 232 “It is important to highlight that the H-square resonance frequency is optimized according to the molecular relaxation time of methane (CH4) ~ 11.5 μs [12].” by “It is important to highlight that the H-square resonance frequency corresponds to an acoustic pressure close to the maximum of pressure generated through the absorption of the modulated laser light by methane molecules. This maximum pressure is correlated to the molecular relaxation time of methane ~ 11.5 μs via the heat production rate [10,15].”
- in line 360 “Additionally, the resonator length is chosen to operate at a frequency, which corresponds to the target gas vibrational-transitional relaxation time.” by “Additionally, the resonator length is chosen to operate at a frequency, which corresponds to an acoustic pressure close to the maximum of pressure generated due to the absorption of the laser light by the target gas.”
- in line 372 “The electrical characterization confirms that the H-square resonator operate at a frequency which corresponds to the molecular relaxation rate of methane, which is 11 kHz.” by “The electrical characterization confirms that the H-square resonator operate at a frequency of 10.28 kHz which corresponds to an acoustic pressure close to the maximum of pressure generated due to the absorption of the laser light by the target gas.”
4. Line 262. "displacement reaches 20 pm". Can you clarify what this value of "pm" is?
Answer: pm=picometers. we add ‘20 pm (picometers)’ in line 286.
5. Figure 6. How can you explain that the maximum of the photoacoustic signal (dark blue and light blue lines) does not coincide with the absorption maximum of methane (black line)?
Answer: The 1f and 2f maxima correspond to the 1st and the 2nd derivative of the absorption peak of methane, respectively (in Figure 6). The maximum absolute value of the 1f signal is obtained when the laser emission wavelength is located in the middle of the edge of the absorption line while the maximum absolute value of the 2f signal is obtained when the laser emission wavelength is located at the same wavelength of the absorption line.
We decided to add line 266 “The 1f and 2f correspond to the 1st and the 2nd derivative of the absorption peak of methane, respectively.”
6. There are some typos, for example line 281, "950Nml/min", should there be a space instead of "N"?
Answer: Indeed, "Nml/min" is a unit of gas flow rate and stands for "normal milliliters per minute". This unit is typically used to express gas flow rate at standard temperature and pressure (STP). STP is defined as a temperature of 0 degrees Celsius (273.15 Kelvin) and a pressure of 1 atmosphere (101.325 kilopascals or 760 millimeters of mercury). It is a commonly used unit in gas flow measurement, particularly for small to moderate gas flow rates.
We added a space between “950” and “Nml/min” line 305.

Round 2
Reviewer 4 Report
The authors answered all questions exhaustively, the article can be published in the presented form.